# Diverse Roles of Mitochondria in Renal Injury from Environmental Toxicants and Therapeutic Drugs

**DOI:** 10.3390/ijms22084172

**Published:** 2021-04-17

**Authors:** Lawrence H. Lash

**Affiliations:** Department of Pharmacology, Wayne State University School of Medicine, Detroit, MI 48201, USA; l.h.lash@wayne.edu

**Keywords:** renal proximal tubule, mitochondria, membrane transport, glutathione, biomarkers, oxidative stress, nephrotoxicity

## Abstract

Mitochondria are well-known to function as the primary sites of ATP synthesis in most mammalian cells, including the renal proximal tubule. Other functions have also been associated with different mitochondrial activities, including the regulation of redox status and the initiation of mitophagy and apoptosis. Mechanisms for the membrane transport of glutathione (GSH) and various GSH-derived metabolites across the mitochondrial inner membrane of renal proximal tubular cells are critical determinants of these functions and may serve as pharmacological targets for potential therapeutic approaches. Specific interactions of reactive intermediates, derived from drug metabolism, with molecular components in mitochondria have been identified as early steps in diverse forms of chemically-induced nephrotoxicity. Applying this key observation, we developed a novel hypothesis regarding the identification of early, sensitive, and specific biomarkers of exposure to nephrotoxicants. The underlying concept is that upon exposure to a diverse array of environmental contaminants, as well as therapeutic drugs whose efficacy is limited by nephrotoxicity, renal mitochondria will release both high- and low-molecular-weight components into the urine or the extracellular medium in an in vitro model. The detection of these components may then serve as indicators of exposure before irreversible renal injury has occurred.

## 1. Introduction

Mitochondria are important organelles in most mammalian cells. Although their most notable function involves oxidative phosphorylation and generation of ATP for cellular work (e.g., active transport, cellular movement, or contraction) and biosynthetic processes, other types of processes that occur in mitochondria play key roles in other diverse types of functions. Examples include mitochondria serving as sensors for oxidation–reduction or redox status and as sensitive and early sites for responses to chemical toxicants. To illustrate the diverse functions of mitochondria, this review focuses on the renal proximal tubular (PT) cell because this is the main target cell population for many environmental toxicants and nephrotoxic drugs. Renal PT cells are among several cell types that are highly dependent on cellular energy, making mitochondria critical to their function. For the renal PT cell, the high demand for ATP is due to their physiological role in electrolyte and fluid balance. Other aspects of physiological and biochemical function also make renal PT cells a perfect exemplar of a cell type for which mitochondrial function and dysfunction underlie a broad array of processes and activities. The review is subdivided into four sections, each focusing on a distinct role or type of role of mitochondria in the renal PT cell.

The first section can be considered more of a classic view of the role of mitochondria in renal PT cellular function because it summarizes the importance of ATP supply. As an example, the response of renal mitochondria to a marked increase in energy demand that occurs during compensatory renal hypertrophy (CRH) after a reduction of functional renal mass after unilateral nephrectomy (NPX) is discussed.

In the second section, the role of renal mitochondria as redox sensors is discussed. Here, the focus is on the underlying importance of redox processes in renal mitochondria and their susceptibility to derangement such that a state of oxidative stress may occur. Subsequently, the regulation and roles of the glutathione system in renal mitochondria are reviewed. Herein, a dual role for reduced glutathione (GSH), as alternately a protective antioxidant and as an activator of certain classes of chemical toxicants, is summarized and illustrated. Additionally, a brief summary of some of the other key redox sensors in mitochondria is provided.

In the third section, the role of renal mitochondria in the sequelae of events underlying chemically-induced nephrotoxicity is briefly reviewed. Here, mitochondria are viewed as important, sensitive, and early target sites for the molecular events that result in decrements in renal PT cell function.

In the final major section, a novel hypothesis of renal mitochondria as a source of biomarkers for exposure to either environmental toxicants or therapeutic drugs whose clinical utility is limited by nephrotoxicity is presented. This hypothesis builds on a large array of data on renal mitochondria as sensitive and early molecular targets of toxicants, as well as many of the unique biochemical properties of these organelles.

## 2. Basic Functions of Renal Mitochondria in Supplying ATP for Cellular Work

A basic principle of bioenergetics is that energy production needs to match energy demand. Different tissues and specific cell types within many tissues have distinctive energy demands based on their physiological and biochemical functions. The kidneys, and the PT cells in particular, are notable in that they have an extremely high energy requirement. This energy requirement, in the form of ATP, has long been known to be predominantly due to the active transport of sodium ions and other cations and anions [1,2,3,4,5,6,7,8]. Importantly, mitochondrial respiratory activity in renal PT cells is tightly regulated by energy demand [8,9].

An illustration of this coupling of energy consumption via active transport to energy supply (i.e., ATP generated primarily by mitochondria) in the renal PT cell is shown in Figure 1. An increase in the plasma concentrations of substrates (e.g., organic anions (OAs), organic cations (OCs), D-glucose, and various D-amino acids) for sodium-linked cotransporters on the basolateral plasma membrane (BLM) should result in the utilization of more of the sodium electrochemical gradient for transport. This would, in turn, stimulate activity of the (Na^+^ + K^+^)-ATPase on the BLM, which would then increase ATP consumption and correspondingly increase mitochondrial ATP production via oxidative phosphorylation.

A more dramatic illustration of the linkage between energy demand and energy production can be found in the example of compensatory renal hypertrophy (CRH) that results from a reduction of renal mass, such as from unilateral nephrectomy (NPX). A reduction of functional renal mass can occur as a result of renal disease, surgery, or aging. Within a very short period of time after a reduction of the number of functional nephrons, the remnant renal tissue undergoes a large array of biochemical, physiological, and morphological changes to compensate for the loss in function [10,11]. Rather than being an increased synthesis of cells (i.e., hyperplasia), the response is predominantly hypertrophy, in which existing cells (predominantly from the renal PT region) increase in size [10,11]. The NPX rat has been frequently used as a model for reduced functional renal mass. Within several hours of the removal of one kidney, the contralateral or remnant kidney exhibits large increases in sodium ion transport and glomerular filtration [10,11,12]. Consistent with this increased workload per unit mass, it has long been known that mitochondria in the remnant renal tissue proliferate [13]. Biochemically, both the activity and expression of the (Na^+^ + K^+^)-stimulated ATPase increase dramatically [14,15]. Moreover, consistent with the concept that mitochondrial respiration increases in response to an increased demand for ATP, renal PT cells in the remnant kidney are described as being in a hypermetabolic state [16,17].

To better understand the biochemical and toxicological implications of CRH, we studied responses of the PT cell from the remnant kidney of NPX rats both in vivo and in vitro in a series of papers, focusing on mitochondrial function and oxidation–reduction (or redox) status [18,19,20,21,22,23,24,25,26,27]. Additionally, we examined the impact of CRH on the susceptibility of PT cells to several nephrotoxicants (e.g., inorganic mercury and tert-butyl hydroperoxide (tBH)), demonstrating that the altered cellular biochemistry and physiology result in increased cytotoxicity. Figure 2 shows the enlarged size and altered morphology of primary cultures of rat PT cells from NPX rats compared to those from normal rats. The figure is meant to provide a qualitative assessment of CRH, in which many, but not all, cells are enlarged. Cytokeratin staining, which is a marker for epithelial cells, was performed to better visualize individual cells.

As shown in Figure 3, the initiating events in the CRH response include an increased glomerular filtration rate (GFR) and an increased fractional sodium excretion (FeNa). These events, in turn, activate the plasma membrane (Na^+^ + K^+^)-ATPase. Similar to the relationships illustrated in Figure 1, the activation of the (Na^+^ + K^+^)-ATPase increases the mitochondrial respiration rate (QO_2_) via increased ATP consumption. This increased respiratory rate results in increases in the release of reactive oxygen species (ROS). It is important to mechanistically distinguish between early responses to increases in ROS and those that occur with prolonged or more marked increases in these oxidant-promoting molecules. As described in the next section and conceptualized by Jones and colleagues [28,29,30,31], early increases in ROS are viewed as having a signaling function in regulating the redox circuitry of the mitochondria and other subcellular compartments. Thus, in PT cells from NPX rats, early increases were observed in activities (but not mRNA or protein expression) of the two mitochondrial transporters for GSH, the dicarboxylate and 2-oxoglutarate carriers (DIC and OGC, respectively), and in intramitochondrial concentrations of GSH [26,27]. At later times, however, mitochondrial GSH concentrations in PT cells from NPX rats have been found to decrease compared to those in PT cells from control rats, and an enhanced injury from a variety of toxicants has been observed [22,23,25].

To further understand the molecular changes underlying the CRH response, we conducted a proteomics analysis of renal tissue from control and NPX rats using the iTRAQ^TM^ (isobaric tags for relative and absolute quantitation) labeling method for quantitative proteomics with tandem mass spectrometry (Benipal, B.; Stemmer, P.M.S.; Lash, L.H. unpublished data) (Table 1).

The iTRAQ^TM^ reagent (from Applied Biosystems; Foster City, CA, USA) labels free amines on peptides with a stable isotope containing an isobaric tag. A portion (100 µg) of each sample was reduced with dithiothreitol, alkylated with iodoacetamide, and digested overnight at 37 °C with trypsin. Each sample was labeled with a unique isobaric iTRAQ^TM^ tag (control/buffer = 114; NPX/Buffer = 115). Samples were combined, purified by strong cation exchange chromatography using the SCX Methods Development Kit (Applied Biosystems), dried, resuspended in water (2 cycles), and then dried and resuspended in 2% acetonitrile and 0.1% trifluoroacetic acid. Peptides were separated by reverse phase chromatography (Magic C18 column, Michrom), ionized with the ADVANCE ion source (Michrom), and introduced into an LTQ-XL mass spectrometer (Thermo Fisher Scientific). Abundant species were fragmented in the pulsed-Q dissociation mode (PQD), with 5 short and 2 long LC/MS/MS runs being performed on each sample. Data analyses were performed using the BioWorks (Thermo) and Scaffold software (Proteome Software), which incorporates the SEQUEST, X!Tandem, and ProteinProphet algorithms. A rat protein database (IPI ver. 3.47) was used for analyses. Using a criterion for significance of 1.5-fold, the majority of proteins that were more highly expressed in kidney from NPX rats were found to be of mitochondrial origin (Table 1).

## 3. Renal Mitochondria as Redox Sensors

### 3.1. Susceptibility to Oxidative Stress

Although we can view mitochondrial function as serving to maintain a balance between ATP demand and ATP supply, there are several factors that are important in the adaptability and plasticity that allow these organelles to respond to changing environmental conditions. While the involved factors or mechanisms, such as electrochemical potential or pH gradients, are considered intrinsic to mitochondrial function, other factors such as redox circuitry are believed to be critical for the ability of mitochondria to respond to physiological, pathological, and toxicological perturbations [28,29,30,31,32,33]. The interplay between bioenergetic processes in mitochondria and the various redox regulatory processes that help regulate metabolism and overall cellular function is illustrated in Figure 4.

Through glycolysis, the citric acid cycle, and the mitochondrial electron transport chain, carbohydrate metabolism generates energy in the form of transmembrane gradients of pH (∆pH), membrane potential (∆ψ), and substrate concentration (∆ (substrate)). These forms of energy can modulate mitochondrial function via one of several types of post-translational modifications. Other forms of post-translational modifications occur through redox regulation linked to the NADPH/NADP^+^ redox couple. In these manners, changes in levels of ROS or reactive nitrogen species (RNS) may result in different levels of post-translationally modified proteins, thus leading to functional changes. An important implication of these links is that excessive levels of any of the key signaling molecules will result in altered protein function, leading to toxicity. Moreover, one must distinguish between physiological variations in levels of the signaling molecules, such as those that might occur during exercise or short-term starvation, and pathological variations that might occur either during exposure to nephrotoxic chemicals or drugs or during pathological disease states such as ischemia-reperfusion or diabetic kidney disease.

### 3.2. The Glutathione System as Both Protective Antioxidant and Activator for Nephrotoxicity

#### 3.2.1. Mitochondrial Glutathione as a Protective Antioxidant

The scheme shown in Figure 4 is obviously an oversimplification and is designed to focus on the key mediators and the overall regulatory processes. Multiple intermediates serve to communicate redox status and do so, in large part, by controlling levels of signaling molecules such as O_2_^−^, H_2_O_2_, H_2_S, and NO. One of the major intermediates is reduced GSH. Though GSH exists in many forms, including as mixed disulfides with proteins and as low-molecular-weight disulfides with L-cysteine or another molecule of GSH, the major form in mitochondria under normal conditions is free GSH. To understand the role of mitochondrial GSH (mtGSH) in redox control and susceptibility to oxidative stress, it is first necessary to delineate how the pool of mtGSH is derived and regulated.

The reader is referred to earlier reviews of renal mtGSH transport for more background and details on the identification of specific carriers involved in the transport process [34,35,36]. The primary rationale for determining the derivation and regulation of mtGSH was based on the realizations that renal GSH synthesis appears to exclusively occur in the cytoplasm [37], that renal mitochondria contain a distinct pool of GSH with concentrations in the millimolar range [38], and that the GSH molecule has a net negative charge at physiological pH. Based on these facts and properties, as well as the presence of numerous carriers in the mitochondrial inner membrane for a diverse array of organic anions and zwitterions (including amino acids) [39], the most likely derivation of the mtGSH pool would be from transport of cytoplasmic GSH into the mitochondrial matrix via one or more electroneutral or electrogenic anion exchange carriers. In a series of studies with isolated renal cortical mitochondria from rat kidney and purified and reconstituted renal mitochondrial anion carriers [37,40,41], we demonstrated that two electroneutral anion exchangers in the renal mitochondrial inner membrane, the dicarboxylate carrier (DIC; *Slc25a10*), and the 2-oxoglutarate carrier (OGC; *Slc25a11*), account for most of the observable transport of GSH from the cytoplasm into the mitochondrial matrix. The function of these two carriers is illustrated in Figure 5.

Although the focus of this review is on renal mtGSH, many other tissues have similarly distinct mtGSH pools, and the function of both the DIC and OGC in transport of GSH from the cytoplasm into mitochondrial matrix has been described in other tissues, including the liver [42,43], lung [44,45], cardiac myocytes [46], colonic epithelium [47,48], and brain [49,50]. Despite the presence of the DIC, OGC, and GSH in all mitochondria, some tissue-specific differences in the transport processes between renal mitochondria and those from other tissues, such as the liver [43], have been noted.

In addition to the obvious role of mtGSH, which is present at millimolar concentrations, in maintaining the reduced state of mitochondrial protein thiols, the mtGSH pool is also closely linked to the metabolic state of the renal PT cell. This can be best appreciated by examining the role of 2-OG in mitochondrial intermediary metabolism and the various cellular GSH transport processes (Figure 6). Studies on the transport of GSH across the BLM from the interstitial space/plasma into the renal PT cell have shown that multiple carriers are involved, including Oat3, (possibly) Oat1, and NaC3 [34,51]. 2-OG serves as a co-substrate for all of these carriers, providing the link with the mitochondrial carriers for GSH.

From the scheme, it should be clear that while mtGSH levels may help maintain mitochondrial redox status and thus mitochondrial function, mitochondrial energetics, which supply ATP through the tricarboxylic acid (TCA) cycle and oxidative phosphorylation (OXPHOS), also supply 2-OG as a transporter co-substrate for multiple carriers on both the mitochondrial inner membrane and the BLM. Hence, alterations in substrate supply, such as those that might occur during starvation, could significantly impact the energetic driving forces for GSH transport.

A clear demonstration of the importance of these mitochondrial anion transporters in maintaining mitochondrial and cellular function was provided by studies conducted in Normal Rat Kidney 52 Epithelial (NRK-52E) cells, an immortalized cell line derived from rat PT cells. Although these cells exhibit some of the limitations of other immortalized cell lines, they exhibit many properties of in vivo renal PT cells with respect to mitochondrial function and GSH status [52]. To assess the impact of different levels of mtGSH transport function in a renal PT cell line, initial rates of GSH uptake into mitochondria and apoptosis caused by two previously well-characterized mitochondrial toxicants, tBH and *S*-(1,2-dichlorovinyl)-L-cysteine (DCVC), were assessed in four types of NRK-52E cells that express varying levels of mtGSH transporter activity (Figure 7).

The transient overexpression of the DIC or the stable overexpression of the OGC was found to result in marked increases in mtGSH uptake, with the overexpression of either transporter resulting in approximately 6-fold increases in the rate of GSH uptake. Most importantly, NRK-52E cells overexpressing either of the carriers exhibited much less apoptosis due to exposure to either of the two well-established chemicals that are mitochondrial and renal toxicants. Consistent with the ability of the mitochondria to accumulate GSH and use it for antioxidant defense, cells that overexpressed the low-activity, double-cysteine mutant of the OGC exhibited slightly less mtGSH uptake activity than even NRK-WT cells but exhibited a high extent of apoptosis from either tBH or DCVC exposure.

#### 3.2.2. Glutathione-Dependent Bioactivation in Renal Mitochondria

The concept that renal GSH in general and mtGSH in particular can have both critical protective and toxic roles in renal cells was recently reviewed [55]. An important group of environmental pollutants, halogenated alkanes and alkenes, are known to specifically elicit nephrotoxicity by GSH-dependent metabolism in the kidneys, by both enzymes in the renal PT cytoplasm and mitochondria [56,57]. The concept and key data behind the seemingly contradictory role of GSH in bioactivation date back more than 50 years to a set of studies on the metabolism and toxicity of DCVC in liver mitochondria by Parker [58] and Stonard and Parker [59,60]. Although these in vitro studies were not conducted in cells or isolated mitochondria from the in vivo target cell, namely the renal PT cell, they established basic mechanisms and metabolic pathways for the bioactivation process that is dependent on GSH.

A variety of halogenated alkanes and alkenes, including trichloroethylene, perchloroethylene, hexachlorobutadiene, chlorotrifluoroethylene, and tetrafluoroethylene, all undergo conjugation with GSH, catalyzed by GSH *S*-transferases, to form GSH *S*-conjugates [56,57]. These GSH *S*-conjugates are primarily formed in the liver, but they can also form in many other tissues and are directed to the kidneys via interorgan translocation pathways (e.g., enterohepatic and renal-hepatic transfer) [61]. While the intact GSH *S*-conjugates may be taken up from the blood side via transport across the BLM [34,55,62], they are typically transported out into the tubular lumen, presumably by Oatp1a1, as shown for GSH in Figure 6. Other GSH S-conjugates that enter the renal tubules via glomerular filtration are rapidly degraded by GGT and DP activities on the BBM to release the corresponding cysteine *S*-conjugate. While some of these cysteine *S*-conjugates are metabolized to either *N*-acetyl-L-cysteine-*S*-conjugates (i.e., mercapturates) in the cytoplasm, some may also be metabolized by one of several cysteine conjugate-beta-lyases (CCBLs) in either the cytoplasm or mitochondria [56,63,64]. These processes are schematically summarized in Figure 8.

Those cysteine *S*-conjugates that are bioactivated within the mitochondria of renal PT cells have been demonstrated to produce a range of adverse effects, including targeting of specific mitochondrial enzymes, specific mitochondrial macromolecules including proteins, lipids, and DNA, or specific mitochondrial processes. There are extensive data on these responses in mitochondria for chlorotrifluoroethyl-L-cysteine [65], DCVC [66,67,68,69,70,71,72,73,74,75], pentachlorobutadienyl-L-cysteine [76,77], and tetrafluoroethyl-L-cysteine [78,79]. In this manner, the mitochondrial and cellular toxicity of these halogenated alkanes and alkenes is dependent on GSH, thus illustrating a role that contrasts with its typical antioxidant and protective roles.

### 3.3. Other Redox Sensors in Renal Mitochondria

Although the loss of mtGSH clearly sensitizes cells to oxidants and many other types of cytotoxic chemicals, the GSH redox system is not the only one that functions to regulate redox status in mitochondria in the kidneys or other tissues. Jones and colleagues [80,81,82] characterized the role of the thioredoxin (Trx)–thioredoxin reductase (TrxR) system in multiple subcellular compartments, including the mitochondria, as an additional component. In a series of reviews [29,30,32,83,84,85], Jones and colleagues further conceptualized the signaling pathways that regulate redox systems, the redox proteome, and cellular energetics. In this redox systems approach, there are three main components that act to modulate protein cysteinyl residues, resulting in altered protein function (Figure 9). While events impacting cellular function and redox status occur in virtually every subcellular compartment and in the extracellular space, the mitochondria are prominent because of their high rate of metabolic activity in a concentrated space and the existence of numerous redox-sensitive targets. Redox modulators include ROS such as H_2_O_2_ and organic peroxides (ROOH), RNS such as NO and peroxynitrite, hypochlorous acid (HOCl), quinones such as ubiquinone, and various thiols and disulfides. 

## 4. Renal Mitochondria in the Sequelae of Chemically-Induced Nephrotoxicity

An important focus in mechanistic toxicology is the determination of the sequence of events that occur upon the exposure of a target cell to a toxicant. Moreover, the identification of the most sensitive molecular targets in an exposure can provide both mechanistic insights and help in the development of therapeutic agents. In cells such as the renal PT cell, mitochondria are prominent targets for chemicals because of the following factors: (1) the high ATP consumption rate, resulting in a high degree of dependence on mitochondrial function; (2) the high density of mitochondria in the cells; (3) the availability of molecular targets (e.g., abundance of protein sulfhydryl groups); (4) the large array of organic anion and organic cation transporters on the inner mitochondrial membrane that can result in the intramitochondrial accumulation of a diverse array of chemicals; and (5) the presence of bioactivation enzymes within the mitochondria that can convert chemicals into reactive and toxic species. The central role of mitochondria in diverse forms of chemically-induced nephrotoxicity, as well as in renal pathological states such as ischemic injury and diabetic nephropathy, was highlighted by a sampling of key review articles published over the past few years [86,87,88,89,90]. The diversity of chemicals and drugs that produce nephrotoxicity that is associated with mitochondrial dysfunction is highlighted in Table 2.

While a discussion of the detailed mechanisms of action of these various chemicals and drugs is beyond the scope of this review, the key message is that for all these diverse agents and the pathological states of hypoxia or ischemia, mitochondrial dysfunction is a prominent, sensitive, and early mode of action. The range of responses and parameters measured in the studies cited in Table 2 include the general assessment of bioenergetic status, the mitochondrial permeability transition, F_o_F_1_-ATPase activity, mitochondrial membrane potential, respiration rate, mitochondrial-mediated apoptosis, mitochondrial Ca^2+^ ion uptake, mitochondrial sulfhydryl status, mitochondrial ROS levels, and mitochondrial transporter function. The diversity of the listed agents and processes, which represent only a sampling and are not exhaustive, should convey the concept that mitochondria play a central role in the response of the renal PT cell to many toxicants, nephrotoxic drugs, and pathological states.

## 5. Renal Mitochondria as a Source of Biomarkers for Chemical Exposure

Besides being prominent targets for many environmental toxicants and drugs, the mammalian kidneys and the renal PT cells in particular often exhibit dysfunction associated with many diseases. Both chronic kidney disease (CKD) and end-stage renal disease (ESRD) are frequently the result of either chronic diseases such as diabetes or exposure to environmental contaminants or therapeutic drugs whose clinical utility is limited by nephrotoxicity [115]. The standard indicators or biomarkers of renal function that are routinely used clinically, namely serum creatinine (SCr) or blood urea nitrogen (BUN), are relatively insensitive and typically do not exhibit noticeable elevations until there is a significant extent of renal injury that is often irreversible. To reduce the morbidity and mortality associated with CKD and ESRD, so-called biomarkers are needed that can indicate exposure or pathological states at a stage at which kidney injury is either minimal or still reversible.

Interest in the identification of biomarkers for kidney function that are based on non-invasive or minimally invasive methods, such as urine collection or blood samples, respectively, has been strong for more than two decades. One of the earliest attempts at systematizing efforts to identify and clinically apply renal biomarkers was a workshop on Biomarkers in Urinary Toxicology convened by the National Research Council, which published a report in 1995 [116]. A recent special issue of *Science* in 2019 [117] highlighted the continued interest in this topic. Over the past two decades or so, a number of additional and more sensitive biomarkers, such as Kidney Injury Molecule-1 (KIM-1), cystatin C, and *N*-acetyl-β-D-glucosaminidase (NAG), have been identified and validated. Indeed, the U.S. Food and Drug Administration (FDA) has developed validation protocols for new biomarkers for multiple target organs including the kidneys [118]. Although many of the new biomarkers for kidney function, such as KIM-1 and cystatin C, are dramatically more sensitive than the standards SCr and BUN, they still possess significant limitations in that they reflect some degree of renal injury and are not necessarily linked to the mechanism of action of specific drugs.

Considering the sensitivity of mitochondria in renal PT cells and the broad array of chemicals and pathological conditions that are associated with mitochondrial perturbations and damage, a hypothesis was proposed that changes in components from renal mitochondria could be detected in either urine or plasma soon after exposure to a broad array of nephrotoxic chemicals or during the early stages of disease progression. These components may include proteins (either altered in abundance or covalently modified), lipids, or low-molecular-weight metabolites.

To test this hypothesis, namely that PT cells primarily release mitochondrial components upon exposure to an array of environmental toxicants or drugs, primary cultures of human proximal tubular (hPT) cells were incubated with either a regular cell culture medium (=control) or DCVC (10, 100, or 250 µM) for either 4 or 24 h. Samples from the extracellular media were collected and analyzed by iTRAQ^TM^ labeling and LC–MS/MS in a manner similar to that in studies described in Table 1. To support the hypothesis that hPT cells exposed to a nephrotoxicant release mitochondrial contents into the extracellular space, the subcellular compartment from which the identified derived proteins were categorized (Table 3).

In the analysis, peptides from 311 proteins were identified, although only 299 had sufficient spectral information to permit quantitation. Although proteins that were detected were derived from almost every subcellular compartment, it is quite notable that mitochondria were by far the most significant contributor of detected proteins. Additional analysis using TMT^TM^ multiplexing and LC–MS/MS to compare proteins found in the extracellular space from control vs. DCVC-treated cells identified several proteins that were differentially expressed. Although not all were of mitochondrial origin, two that exhibited significant dose- and exposure time-dependent increases are illustrated in Figure 10. The results showed two potential mitochondrial enzymes that are released from exposed hPT cells and may thus serve as early biomarkers of exposure. These studies were preliminary. Similar studies need to be conducted with other toxicants and drugs to demonstrate the broad applicability of the hypothesis.

## 6. Summary and Conclusions

This review sought to summarize some of the diverse roles of mitochondria in renal PT cellular function. These diverse roles include classic functions of mitochondria as the major source of ATP for the myriad of energy-dependent functions and as prominent target sites for toxic effects in the early sequence of events after chemical exposure, including initiating apoptosis. Additionally, mitochondria play important roles in the regulation of redox status in the cell by using a network of distinct systems, including the GSH and Trx systems. Though GSH is traditionally viewed as a prominent intracellular and intramitochondrial antioxidant, GSH also plays a role in the bioactivation and nephrotoxicity of a group of halogenated alkanes and alkenes. While the key bioactivation step occurs in both the cytoplasm and mitochondria, the mitochondria are especially susceptible to toxic effects from bioactivated cysteine conjugates. Finally, a novel role for mitochondria—providing a source of sensitive biomarkers of exposure to a diverse array of environmental toxicants and therapeutic drugs whose clinical use is dose-limited by nephrotoxicity—was discussed. Support for this hypothesized role is based on some preliminary proteomics data and is a logical extension of some of the other roles for mitochondria.

## Figures and Tables

**Figure 1 ijms-22-04172-f001:**
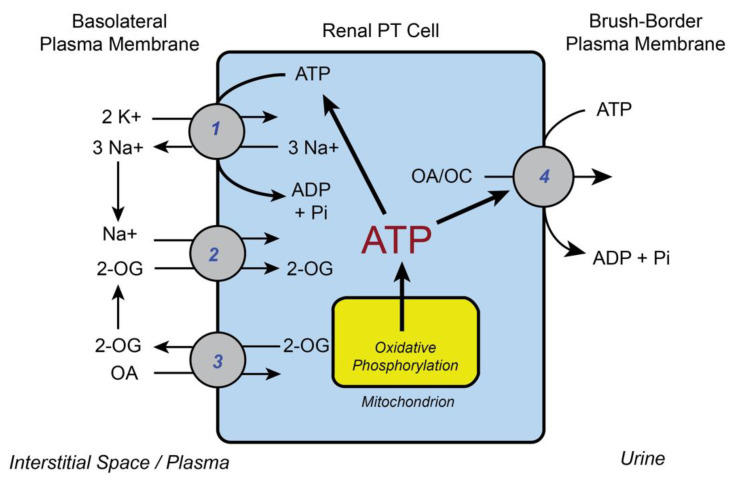
Generalized scheme showing coupling of various active transport processes in the renal proximal tubular (PT) cell to ATP levels and mitochondrial function. 1: (Na^+^ + K^+^)-stimulated ATPase; 2: sodium-dicarboxylate 3 cotransporter (NaC3; *Slc13a3);* 3: organic anion transporter 1 or 3 (Oat 1/3; *Slc22a6/8*); and 4: multidrug resistance-associated protein 2 or 4 (Mrp2/4; *Abcc2/4*) or P-glycoprotein (Mdr1; *Abcb1*). Abbreviations: OA: organic anion; OC: organic cation; 2-OG: 2-oxoglutarate; Pi: inorganic phosphate.

**Figure 2 ijms-22-04172-f002:**
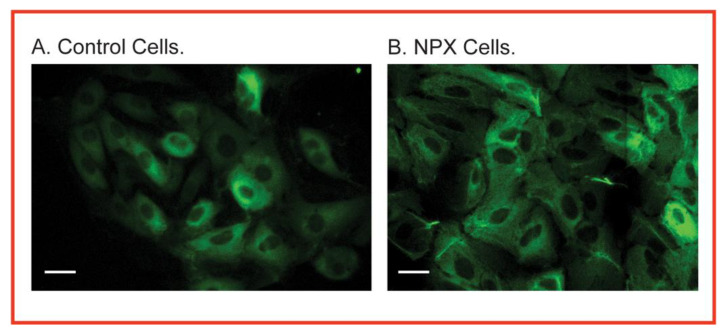
Immunofluorescent staining for cytokeratins in renal PT cells from control and NPX rats on day 5 of primary culture. Freshly isolated PT cells from control (**left**) or NPX (**right**) rats were seeded at a density of 0.5–1.0 × 10^6^ cells/mL and allowed to grow for 5 days. Cells were then washed twice with sterile phosphate-buffered saline, and cytokeratin expression was visualized using a monoclonal fluorescein isothiocyanate-conjugated anti-mouse cytokeratin antibody. Note the enlarged size of many of the PT cells from the NPX rats relative to that of the PT cells from the control rats. Photomicrographs were taken at 100X magnification on a Zeiss confocal laser microscope. Bar = 5 mm. Adapted from [22].

**Figure 3 ijms-22-04172-f003:**
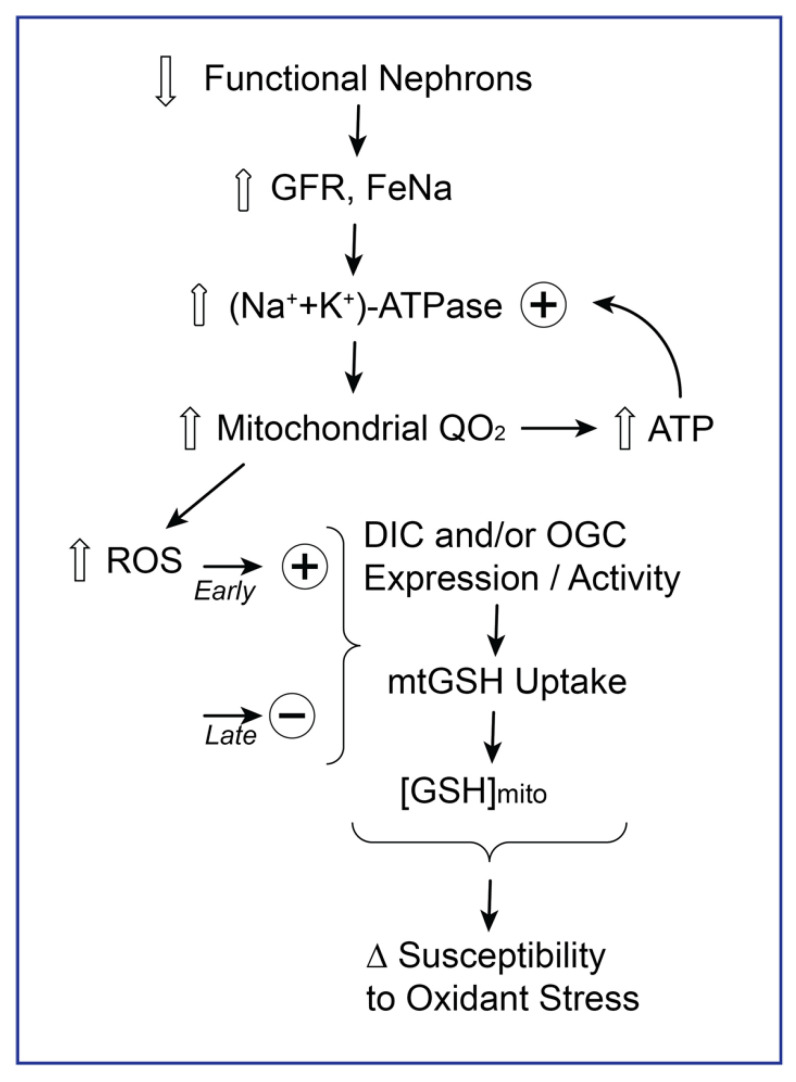
Summary scheme showing impact of compensatory renal hypertrophy (CRH) on renal PT cell mitochondrial function and redox status. The initial compensatory response to a loss in the number of functional nephrons leads to an increase in the glomerular filtration rate (GFR) and fractional sodium excretion (FeNa). These changes, in turn, activate the plasma membrane (Na^+^ + K^+^)-ATPase, which increases cellular ATP consumption. As a consequence, the mitochondrial respiration rate (QO_2_) increases. The persistence of these changes ultimately results in increased levels of reactive oxygen species (ROS), which impact mitochondrial redox status. Initially, compensatory increases are observed in antioxidants, such as glutathione (GSH). However, with continued hypermetabolism in the mitochondria, oxidant stress and increased cell injury occurs. + = stimulated; − = inhibited.

**Figure 4 ijms-22-04172-f004:**
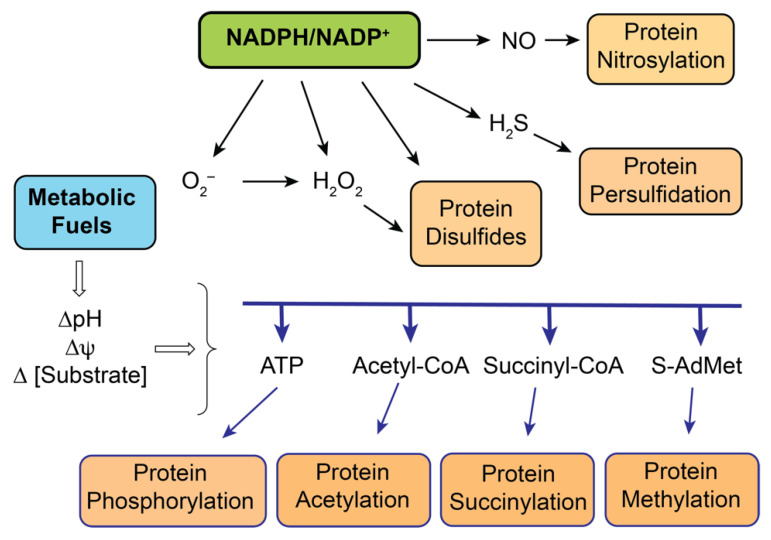
Involvement of mitochondrial metabolism and redox processes in the post-translational regulation of mitochondrial proteins and enzymes. The scheme illustrates the involvement of redox processes (acting through NADPH/NADP^+^) and metabolic processes to produce several types of protein modifications that can alter protein structure and/or function. Abbreviations: S-AdMet: S-adenosylmethionine; NO: nitric oxide. Adapted from [29].

**Figure 5 ijms-22-04172-f005:**
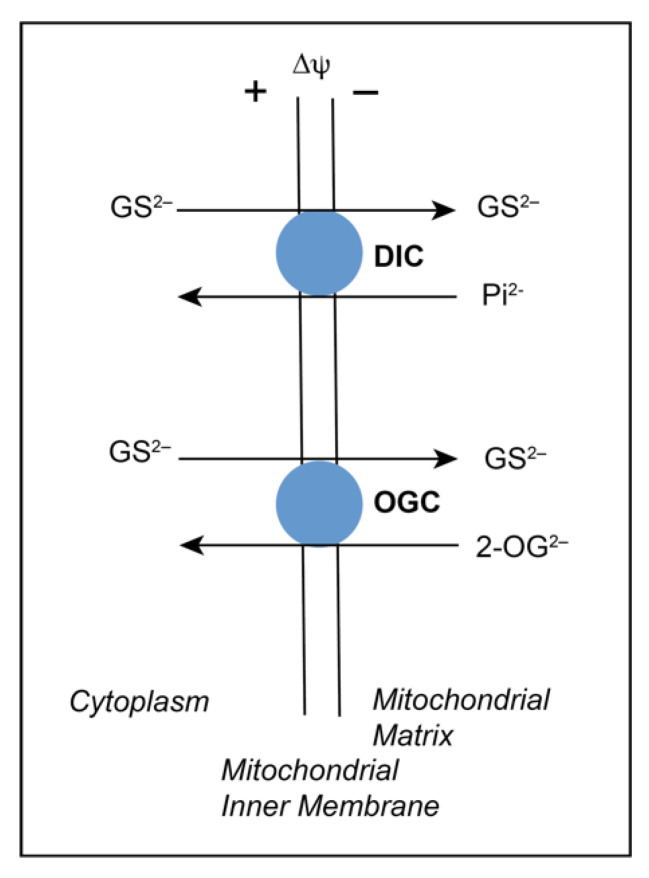
Transport of GSH into renal mitochondria via the dicarboxylate carrier (DIC) and the 2-oxoglutarate carrier (OGC). The scheme illustrates the function of two key inner membrane anion exchange carriers in the movement of GSH from the cytoplasm into the mitochondrial matrix. The DIC catalyzes electroneutral exchange of inorganic phosphate (Pi^2−^) with various organic anions such as malate. The OGC catalyzes the electroneutral exchange of 2-oxoglutarate (2-OG^2–^) with other dicarboxylates such as malate. The GSH molecule is a zwitterion containing a free amino group (positive at physiological pH), two free carboxyl groups (negative at physiological pH), and a sulfhydryl or thiol group (-SH or -S^–^). The last group has a pKa of 8.3, so it is extensively deprotonated at the modestly alkaline pH of the mitochondrial matrix. In this manner, GSH is primarily transported as a dianion in electroneutral exchange for either Pi^2–^ or 2-OG^2–^.

**Figure 6 ijms-22-04172-f006:**
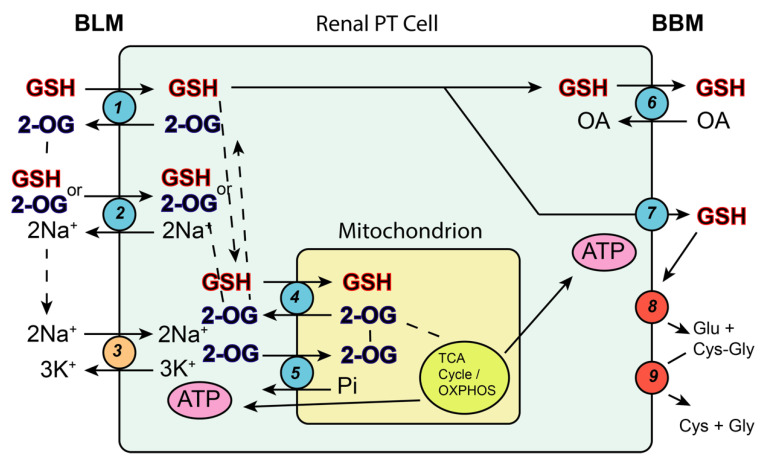
Pathways of plasma membrane and mitochondrial GSH transport in the renal PT cell. The scheme shows the major membrane carriers for GSH on the mitochondrial inner membrane and on the basolateral membrane (BLM) and the brush-border membrane (BBM). The integration of these carriers with a supply of 2-oxoglutarate and a supply of ATP and Na^+^ ion gradients is highlighted with dashed lines. A critical role for mitochondrial energetics, which supplies the PT cell with ATP via the tricarboxylic acid (TCA) cycle and oxidative phosphorylation (OXPHOS), is also evident. Processes and their functions: (1) Organic anion transporter 1 or 3 (Oat1/3; *Slc22a6/8*): transports GSH in exchange for 2-OG at the BLM; (2) sodium-dicarboxylate carrier (NaC3; *Slc13a3*): transports 2-OG in exchange for 2 Na^+^ ions at the BLM and provides 2-OG for Oat1/3; (3) (Na^+^ + K^+^)-ATPase: establishes Na^+^ and K^+^ ion gradients across the BLM and provides Na^+^ as a driving force for many Na^+^-coupled transporters; (4) 2-oxoglutarate carrier (OGC; *Slc25a11*): transports GSH in exchange for 2-OG across the mitochondrial inner membrane; (5) dicarboxylate carrier (DIC; *Slc25a10*): transports either 2-OG or GSH in exchange for inorganic phosphate (Pi) across the mitochondrial inner membrane; (6) organic anion transporting polypeptide 1a1 (Oatp1a1; *Slco1a1*): transports GSH across the BBM and into the tubular lumen in exchange for various organic anions (OA); (7) multidrug resistance protein 2/4 (Mrp2/4; *Abcc2/4*): primary active transporters (ATP-dependent) that catalyze efflux of GSH across the BBM and into the tubular lumen; (8) γ-glutamyltransferase (GGT): enzyme on BBM that cleave the γ-glutamyl residue from GSH; (9) cysteinylglycine dipeptidase (DP): enzyme on BBM that cleaves L-Cys-Gly into constituent amino acids.

**Figure 7 ijms-22-04172-f007:**
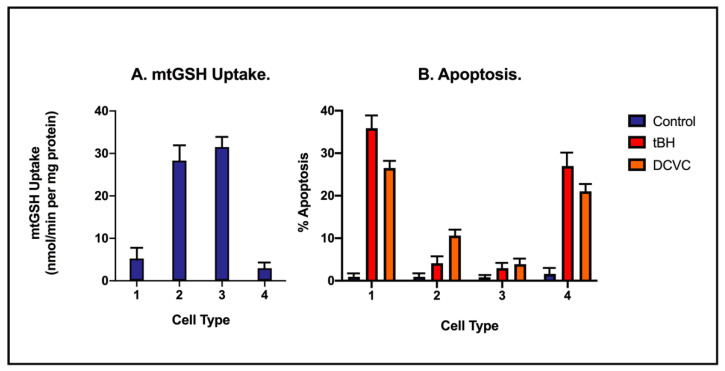
Impact of genetic modulation of mtGSH transporter expression in NRK-52E cells on mtGSH uptake and apoptosis caused by two mitochondrial toxicants. The cDNA for the rat kidney DIC was transiently overexpressed in NRK-52E cells (NRK-DIC = cell type 2), and the cDNA for either the normal OGC (NRK-OGC = cell type 3) and that of a double-cysteine mutant OGC with ~20% of normal activity (NRK-OGC-C221,224S = cell type 4) were stably expressed. Responses were compared to those of wild-type NRK-52E cells (=cell type 1). (**A**) Initial uptake rates for mtGSH. Mitochondria were isolated from each cell population and were incubated with [^3^H]-GSH (final concentration = 5 mM). Data are means ± SD of initial uptake rates from 3–5 separate experiments. (**B**) Percent of apoptotic cells. The four different cell populations were incubated for 4 hr with a normal cell culture medium (=control), 10 µM tBH, or 50 µM DCVC. The percentage of cells undergoing apoptosis (sub-G_1_ fraction) was estimated by propidium iodide staining, flow cytometry, and fluorescence-activated cell sorter (FACS) analysis. Results are means ± SD of 4–5 separate experiments. While the data were derived from previously published primary studies [53,54], the combined summary figure was adapted and modified from a previous review paper [35].

**Figure 8 ijms-22-04172-f008:**
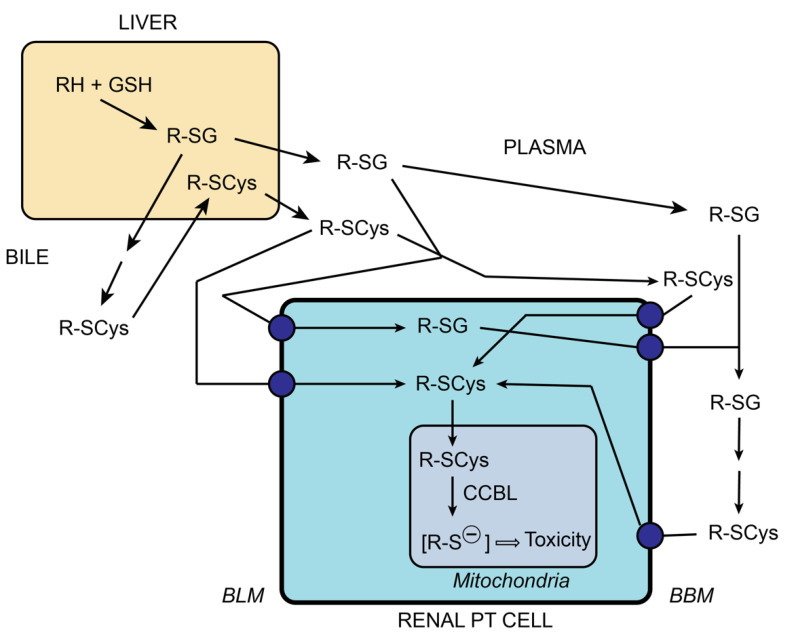
Interorgan metabolism of nephrotoxic GSH and cysteine *S*-conjugates. The scheme emphasizes the roles of initial GSH conjugation, which primarily occurs in the liver, and interorgan transport pathways that deliver either the GSH *S*-conjugate or the cysteine *S*-conjugate to the kidneys, which rapidly transport these conjugates into the PT cell, where the cysteine *S*-conjugate can be bioactivated by the cysteine conjugate beta-lyase (CCBL) within mitochondria to produce a reactive thiolate or related metabolites (e.g., [R-S^–^]) that cause toxicity.

**Figure 9 ijms-22-04172-f009:**
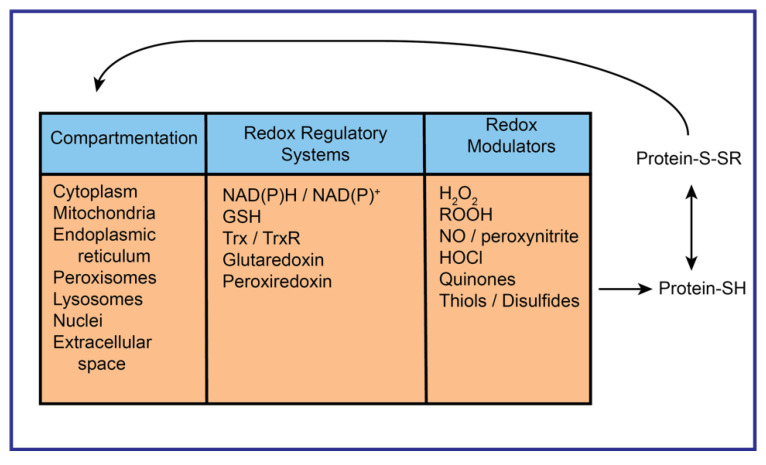
A systems biology approach to redox regulation. The three primary components of the system are subcellular compartments and the extracellular space, several redox regulatory systems, and redox modulators or signaling molecules. Within specific compartments, changes in activities of redox regulatory systems lead to changes in the concentrations of redox modulators. These, in turn, alter the redox state of protein-cysteinyl residues, which, in turn, impact function in the different cellular or subcellular compartments. Abbreviations: GSH: reduced glutathione; NO: nitric oxide; ROOH: organic peroxide; Trx: thioredoxin; TrxR: thioredoxin reductase.

**Figure 10 ijms-22-04172-f010:**
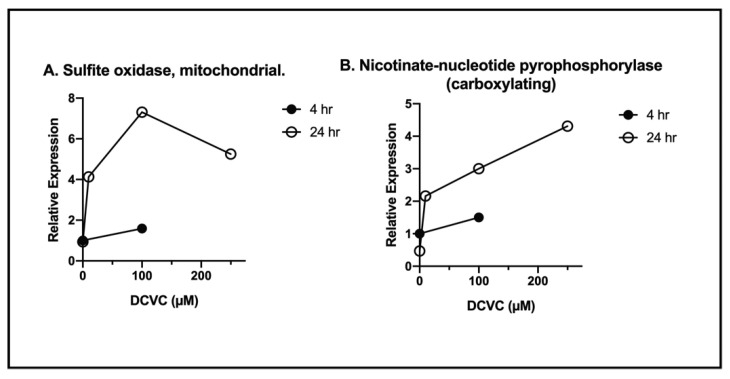
Two mitochondrial proteins identified in culture media of human proximal tubular (hPT) cells exposure to *S*-(1,2-dichlorovinyl)-L-cysteine (DCVC) (10, 100, or 250 µM) for 4 or 24 h. Primary cultures of hPT cells were incubated with either a culture medium (=control) or DCVC, and samples from the extracellular fluid were analyzed using TMT^TM^ multiplexing and LC–MS/MS analysis, as described in the legend of Table 3. Results are shown as expression levels relative to that in the 4-h control samples and are from a single set of analyses.

**Table 1 ijms-22-04172-t001:** Summary of selected proteomics data of mitochondria from control and NPX rat kidneys using iTRAQ^TM^ labeling and liquid chromatography/mass spectrometry/mass spectrometry (LC/MS/MS) analysis. Results are the ratios of protein abundance relative to a control and are means ± SD of 5 sample LC/MS/MS runs. Fold changes in ratios of NPX: control ≥ 1.50 are highlighted in bold.

Accession No.	Classification/Protein	NPX/Control (Mean ± SD)
	ATP Synthesis, Transport and Metabolic Processes	
P15999	Atp5a1 ATP synthase F1 subunit alpha precursor, mitochondrial	**1.91 ± 0.05**
P10719	Atp5b ATP synthase F1 subunit beta precursor, mitochondrial	1.33 ± 0.71
P05141	ADP/ATP translocase 2, Slc25a4	1.03 ± 0.36
	**Tricarboxylic Acid Cycle (Citrate Cycle)**	
P56574	Isocitrate dehydrogenase (NADP) (oxalosuccinate decarboxylase)	**1.88 ± 0.56**
Q9ER34	Aconitate hydratase (aconitase)	**2.54 ± 0.32**
	**Respiratory Electron Transport Chain**	
P13803	Electron transfer flavoprotein subunit alpha	1.00 ± 0.11
Q66HF1	NADH-ubiquinone oxidoreductase 75 kDa subunit	**1.90 ± 0.20**
	**Amino Acid and Fatty Acid Metabolic Pathways**	
P50442	Glycine amidotransferase (transamidinase)	**1.92 ± 0.15**
Q64565	Alanine-glyoxylate aminotransferase 2 (beta-alanine-pyruvate aminotransferase)	**2.13 ± 0.15**
Q02253	Methylmalonate-semialdehyde dehydrogenase (malonate-semialdehyde dehydrogenase)	1.32 ± 0.31
P08503	Medium-chain specific acyl-CoA dehydrogenase	**2.62 ± 0.35**
Q64428	Trifunctional enzyme subunit alpha (3-hydroxyacyl-CoA dehydrogenase)	0.70 ± 0.22
P10860	Glutamate dehydrogenase 1	**1.94 ± 0.41**
	**Anion Transport (Transmembrane Transport)**	
Q9Z2L0	Voltage-dependent anion-selective channel protein 1 (VDAC-1)	1.12 ± 0.08
	**Oxidoreductase Activity**	
Q6AYT0	Quinone oxidoreductase (zeta-crystallin)	1.21 ± 0.23
Q07523	Hydroxyacid oxidase 2	1.47 ± 0.74
Q68FT3	Pyridine nucleotide-disulfide oxidoreductase domain-containing protein 2	**1.92 ± 0.22**

**Table 2 ijms-22-04172-t002:** Selective list of chemicals and drugs whose nephrotoxicity is associated with mitochondrial dysfunction.

Chemical/Drug or Pathological State	Classification	References
DCVC	Cysteine conjugate of environmental pollutant trichloroethylene	[54,62,63,69,70,72,73,74,78]
PCBC	Cysteine conjugate of occupational contaminant hexachloro-butadiene	[76,77]
Bromohydroquinone and its cysteine conjugate	Cysteine conjugate of chemical synthetic intermediate bromohydroquinone	[91,92]
Cephalosporins	Antibiotics	[93,94]
Gentamicin	Antibiotic	[95,96,97]
Tenofovir	Antiviral	[98,99,100,101,102]
Cisplatin	Chemotherapeutic agent	[103,104,105,106,107]
Oxalate	Metabolite	[108]
HgCl_2_	Environmental contaminant	[109]
Hypoxia/ischemia	Pathological states of oxygen deprivation with or without blood flow disruption	[110,111,112,113,114]

Abbreviations: DCVC: *S*-(1,2-dichlorovinyl)-L-cysteine; PCBC: pentachlorobutadienyl-L-cysteine.

**Table 3 ijms-22-04172-t003:** Distribution of proteins in subcellular compartments from culture media of human proximal tubular (hPT) cells incubated with *S*-(1,2-dichlorovinyl)-L-cysteine (DCVC). Data are shown for primary cultures of hPT cells incubated with 250 µM DCVC for 24 h. A portion of each sample of extracellular medium was reduced with dithiothreitol, alkylated with iodoacetamide, and digested overnight with trypsin. Samples were then labeled with a unique isobaric TMP^TM^ reporter tag. Multiplexed samples were analyzed using a Q Exactive Orbitrap mass spectrometer (Thermo Fisher Scientific). Abundant species were fragmented in high-energy collision dissociation mode. Data were analyzed using Proteome Discover version 1.4.0.288 (Thermo) and Scaffold Q+ version 4.1.1 (Proteome Software), which incorporate the Sequest, X! Tandem, and ProteinProphet algorithms. The SwissProt_2013_04 database with 20,253 entries was used.

Subcellular Compartment	# Proteins Identified	% Total
Mitochondria	118	43.2
Cytosol	30	11.0
Vesicles	21	7.7
Soluble	18	6.6
Vacuoles	13	4.8
Lysosomes	12	4.4
Melanosomes	12	4.4
Cell Surface	11	4.0
Microsomes	9	3.3
Nucleoid	9	3.3
Vesicular	9	3.3
Outer membrane	6	2.2
Peroxisomes	5	1.8

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
