# Peer review of "Diverse Roles of Mitochondria in Renal Injury from Environmental Toxicants and Therapeutic Drugs"

_ijms, 2021, doi:10.3390/ijms22084172_

Round 1

Reviewer 1 Report

Dear Author,

I am appreciated to read your work. It is clearly visible that you have wide knowledge and topic is very well elaborated. The great number of cited literature guaranties high quality of presented studies. I suggest to accept manuscript in present form.

Reviewed manuscript have important value for further studies. The wide spread analysis of accessible literature is a good point for planning of studies on the role of specific biomarkers in nephrotoxicity. Kidneys and renal metabolism is the crucial point of drugs and toxins metabolism and its evacuation. Therefore the monitoring of metabolic status of the proximal tubule cells seems to be a good tool for the stage of nephrotoxicity estimation and chosen drugs (used in therapy) evaluation as more or less for patients. These studies can include both therapeutic drugs and environmental contaminants. Novelty of this work is also the attempt for potential description of renal mitochondria in drug neutralization and possible use of biomarkers in practical use. Renal toxic injury is the dangerous status of an organism leading to intoxication, changes in other organs and finally death. Potential monitoring of biomarkers of cell metabolism in status of intoxication or standard therapy can be useful. I want to stress that reviewed work is a valuable attachment for modern science and can be a good source of compiled knowledge in this scientific area.

Author Response

Response: I thank the reviewer for their assessment of the work.

Reviewer 2 Report

Manuscript number: Int. J. Mol. Sci. 2021, 22, x. https://doi.org/10.3390/xxxxx

The author describes the role and function of mitochondria in renal proximal tubular cells according to renal injury. Because of their diverse roles in the cell, mitochondria may serve as targets for therapeutic approaches. The author describes a hypothesis regarding the identification of biomarkers of exposure to nephrotoxicants.

The review gives a summary and a relevant contribution to latest publications. The review is written very clearly with a variety of examples.

The review is well structured and subdivided in 4 sections. The author describes in detail the role of mitochondria in renal PT cells, the role of mitochondria as redox sensors, the role of mitochondria during nephrotoxicity and the role of mitochondria as biomarkers.

In the first part, the author describes the role of mitochondria in renal proximal tubular (PT) cells. Figure 2 shows rat PT cells from rats with unilateral nephrectomy (NPX) in comparison to PT cells from normal rats. The changes in size, morphology and intensity of cytokeratin are not obvious for me. How did you quantified these changes? What is the conclusion of a greater intensity of cytokeratin?

Please label the renal PT cell and mitochondria in Figure 6.

Please show the results in Figure 7 with all individual values and no bar graphs and the standard deviation.

Figure 8 appears confusing because of angular and crossing lines. It would be better to color the liver and renal PT cell with different colors.

The listing of chemicals and drugs in Table 2 would be clearer, if the chemicals and drugs are listed one below the other. Hypoxia/Ischemia is not a chemical or a drug.

All in all the review is mostly focusing on previous work published by the author and a mixture from previous results (original data) and concepts on mitochondrial functions. The figures should be revised as they are oversimplified. The text may be shortened significantly (about 40%).

Author Response

Reviewer 2:

The author describes the role and function of mitochondria in renal proximal tubular cells according to renal injury. Because of their diverse roles in the cell, mitochondria may serve as targets for therapeutic approaches. The author describes a hypothesis regarding the identification of biomarkers of exposure to nephrotoxicants.

The review gives a summary and a relevant contribution to latest publications. The review is written very clearly with a variety of examples.

The review is well structured and subdivided in 4 sections. The author describes in detail the role of mitochondria in renal PT cells, the role of mitochondria as redox sensors, the role of mitochondria during nephrotoxicity and the role of mitochondria as biomarkers.

In the first part, the author describes the role of mitochondria in renal proximal tubular (PT) cells. Figure 2 shows rat PT cells from rats with unilateral nephrectomy (NPX) in comparison to PT cells from normal rats. The changes in size, morphology and intensity of cytokeratin are not obvious for me. How did you quantify these changes? What is the conclusion of a greater intensity of cytokeratin?

Response: This figure is meant to be a qualitative assessment of general cellular morphology and a basic illustration of how the cells from control and NPX appear. Additional text is included to better explain the features illustrated in the figure at the bottom of page 3. The intensity of cytokeratin staining does not necessarily differ between the control and NPX PT cells. The statement denoting this feature has been deleted from the figure legend. Cytokeratins are markers for epithelial cells and the staining is merely intended to better indicate the individual cells.

Please label the renal PT cell and mitochondria in Figure 6.

Response: The figure has been modified as suggested.

Please show the results in Figure 7 with all individual values and no bar graphs and the standard deviation.

Response: SD values have been added to the values shown in the bar graphs. It is unclear to me what else the reviewer is requesting. Why would all individual values be shown? I disagree that the bar graphs lack clarity and request that the figure remain as presented with the added error bars.

Figure 8 appears confusing because of angular and crossing lines. It would be better to color the liver and renal PT cell with different colors.

Response: The cells are colored as requested. I thank the reviewer for this suggestion.

The listing of chemicals and drugs in Table 2 would be clearer, if the chemicals and drugs are listed one below the other. Hypoxia/Ischemia is not a chemical or a drug.

Response: Because of the desire to include hypoxia/ischemia, the subheading “Chemical / Drug” has been modified to include pathological states. The Table has been reformatted as suggested.

All in all the review is mostly focusing on previous work published by the author and a mixture from previous results (original data) and concepts on mitochondrial functions. The figures should be revised as they are oversimplified. The text may be shortened significantly (about 40%).

Response: This comment is completely unclear. The figures noted above in previous comments have been revised according to the reviewer’s excellent suggestions. Otherwise, I do not see the need to markedly alter the text beyond additional proofreading.

Reviewer 3 Report

The authors have clearly described about the diverse roles of mitochondria in the renal injury from the environmental toxicants and therapeutic drugs. I ask the authors to resubmit it after English Editing by native speakers.

Author Response

Reviewer 3:

The authors have clearly described about the diverse roles of mitochondria in the renal injury from the environmental toxicants and therapeutic drugs. I ask the authors to resubmit it after English Editing by native speakers.

Response: The manuscript has undergone additional proofreading.

Round 2

Reviewer 2 Report

The author has addressed most of my concerns. Therefore I suggest acceptance of the manuscript.